# Calibrating Range Measurements of Lidars Using Fixed Landmarks in Unknown Positions

**DOI:** 10.3390/s21010155

**Published:** 2020-12-29

**Authors:** Anas Alhashimi, Martin Magnusson, Steffi Knorn, Damiano Varagnolo

**Affiliations:** 1Computer Engineering Department, University of Baghdad, Baghdad 10071, Iraq; 2Center for Applied Autonomous Sensor Systems (AASS), Örebro University, 70182 Örebro, Sweden; martin.magnusson@oru.se; 3Department of Autonomous Systems, Otto-von-Guericke University, 39106 Magdeburg, Germany; steffi.knorn@ovgu.de; 4Department of Engineering Cybernetics, Norwegian University of Science and Technology, 7491 Trondheim, Norway; damiano.varagnolo@ntnu.no

**Keywords:** lidar, sensor calibration, heteroskedastic, landmark position estimation

## Abstract

We consider the problem of calibrating range measurements of a Light Detection and Ranging (lidar) sensor that is dealing with the sensor nonlinearity and heteroskedastic, range-dependent, measurement error. We solved the calibration problem without using additional hardware, but rather exploiting assumptions on the environment surrounding the sensor during the calibration procedure. More specifically we consider the assumption of calibrating the sensor by placing it in an environment so that its measurements lie in a 2D plane that is parallel to the ground. Then, its measurements come from fixed objects that develop orthogonally w.r.t. the ground, so that they may be considered as fixed points in an inertial reference frame. Moreover, we consider the intuition that moving the distance sensor within this environment implies that its measurements should be such that the relative distances and angles among the fixed points above remain the same. We thus exploit this intuition to cast the sensor calibration problem as making its measurements comply with this assumption that “fixed features shall have fixed relative distances and angles”. The resulting calibration procedure does thus not need to use additional (typically expensive) equipment, nor deploy special hardware. As for the proposed estimation strategies, from a mathematical perspective we consider models that lead to analytically solvable equations, so to enable deployment in embedded systems. Besides proposing the estimators we moreover analyze their statistical performance both in simulation and with field tests. We report the dependency of the MSE performance of the calibration procedure as a function of the sensor noise levels, and observe that in field tests the approach can lead to a tenfold improvement in the accuracy of the raw measurements.

## 1. Introduction

Localization is essential for applications where robots shall move precisely in the surroundings, and is typically performed by leveraging on measurements acquired through distance sensors. Calibrating these distance measurement sensors so that their readings are as accurate as possible is thus a preliminary task that is essential for achieving good navigation performance at a later stage.

To make a practical and typical (but not exhaustive) example, calibrating a distance sensor may mean considering a measurement model of the type
(1)$$r=f_{\mathrm{bias}}\left(d\right)+f_{\mathrm{st}.\mathrm{dev}}\left(d\right)e$$
where *r* is the noisy sensor reading, *d* is the true distance, $f_{\mathrm{bias}}\left(\cdot \right)$ includes the true distance plus an unknown bias term, and $f_{\mathrm{st}.\mathrm{dev}}\left(\cdot \right)$ is a factor modulating the measurement noise whose stochasticity is induced by a random variable *e* typically assumed standard Gaussian and independent and identically distributed (iid). Such a model is motivated by practical real-life situations, such as the one depicted in Figure 1, and the uncertainty constructed according to the work in [1]. A strategy for calibrating a model such as (Equation 1) would in this case correspond to estimating $f_{\mathrm{bias}}$ and $f_{\mathrm{st}.\mathrm{dev}}$ so that these factors can be accounted for when postprocessing new measurements from the sensor.

The operation of calibrating a sensor is typically performed by comparing the raw measurements of the sensor against readings from an external and sufficiently more accurate system (e.g., a motion capture system) that is considered as ground truth (e.g., as in [2]). Acquiring this ground truth (and thus this external and sufficiently more accurate system), however, may be expensive and time-consuming. It may thus be beneficial to find strategies that substitute this information acquisition step with more easily implementable and cheaper approaches.

For example, this substitution can be performed as follows: assume that in structured environments certain structures do not move (e.g., walls, doors, and corners in the built environment trunks of trees in a forest, etc.). Assume moreover that these structures may produce specific and easily recognizable signatures in the readings (e.g., trunks of trees in a forest produce somehow circular shaped features, as in Figure 2).

As soon as the structures do not move, these signatures may be considered as fixed points in an inertial reference frame. If a distance sensor is moved within this environment, then the measurements from the sensor referring to these fixed points should be such that the relative distances and angles among these fixed points remain the same. The calibration process may then be cast, from an intuitive perspective, as finding models like (Equation 1) for which the measurement process complies with the assumption that “fixed features shall have fixed relative distances and angles”.

The goal of this paper is thus *to understand how to leverage these assumptions on the structure of the environment surrounding the distance sensors for the purpose of building statistically accurate distance sensors calibration strategies.*

To do so we will thus consider using a simple strategy: (a) place some artificial landmarks (i.e., some poles) in random positions in space; (b) calibrate the sensor by making its measurements comply with the fixed-world assumption above.

### 1.1. Literature Review

The strategy described in the previous subsection relates to the existing literature as in the following. First of all, distance sensors are often used for reconstructing environmental maps used by robots to move without colliding with obstacles, as, for example, in [3] (We also note that map generation is not the only application; for example, forestry applications use lidars to measure and monitor the growth of forests, compute trunk’s diameters, and calculate the density of trunks or canopies [4,5]). Several strategies have been proposed to improve distance sensor performance and accuracy through statistical manipulation of their measurement processes. For example, the statistical sensor model for ultrasound sensors was presented in [6] with calibration algorithms in [7] and a good review on odometer calibration is presented in [8]. As a generic definition, statistical sensor calibration is the process of improving sensor accuracy and/or precision through transforming the measurements into something closer to ground truth via combining information about the same sensed quantity that is obtained using different sources of information like ground truth data. Sensor calibration for installation error has been well studied in the literature and, in general, is solved using nonlinear optimization, see for example in [9]. However, in this research, we are not considering the extrinsic calibration of the sensor but we focus on the intrinsic calibration that is dealing with the sensor nonlinearity and heteroskedastic, range-dependent, measurement error. Unfortunately, ground truth is not always available and in some cases, if exists, it is very expensive. Therefore, it is usually substituted with one of the following strategies.

Certain assumptions about the sensor movement and about the surrounding environment, in which the calibration process is shaped as joint parameters and state estimation, for example, lidar calibration from linear motion [10].Another strategy for substituting ground truth information with some other information is to implement appropriate sensor fusion strategies, i.e., to combine redundant information from independent distance sensors. Such a strategy has been used in [10,11], where approximated Expectation Maximization (EM) procedures (in the former) and Markov chain Monte Carlo (MCMC) techniques under Bayesian frameworks (in the later) are used for joint parameter and state estimation combining information from lidars, odometry, and ultrasound sensors. Calibrating the intrinsic parameters of one beam based on other beams of rotating multi-beam lidar attracted large amount of research, for example, in [12,13,14,15]. We note that sensor fusion is a vast topic and there are many publications on calibrating other sensors, for example, magnetometer calibration using inertial sensors in [16], camera and IMU calibration in [17], and lidar and camera calibration in [18]. However, here, we are interested only in calibration that is related to lidars.The last strategy is to use assumptions on the environment, for example, odometer calibration with localization [19]. Another example is to use the planar feature in the environment to calibrate lidars. Originally plane-based calibration was presented for calibrating airborne lidars in [20,21]. Then, the authors of [22] introduced a mathematical model and static calibration for the Velodyne HDL-64E lidar using planar feature and least squares solution. The authors of [23] calibrated a 3D lidar for both the geometric and temporal parameters based on Rényi Quadratic Entropy to formulate an optimization problem that maximizes the quality of the point cloud.

As said above, here we specifically investigate how to substitute ground truth information with assumptions on the environment. Our strategy will intrinsically require localizing (in a sense to be specified later) the position of the sensor within the surrounding environment. This means that our paper relates to the existing literature on localizing sensors in space using noisy measurements of distances from landmarks or beacons. To the best of our knowledge, it is possible to do so using three different approaches:triangulation, where the position is determined through measuring the *angles* between the sensing device and the known landmarks (see, e.g., in [24]);trilateration, where the position is determined through measuring the *distances* between the device and the landmarks (see, e.g., in [25,26]);triangulateration, a strategy that combines both of the above (see, e.g., in [27,28]).

Generally, most algorithms use either *triangulation* or *trilateration* alone, as they require less information from the sensor (measuring both distances and angles, indeed, normally requires more expensive hardware). For this reason several studies on how to localize the position of a sensor using landmarks or beacons mostly use triangulation or trilateration approaches. For example, the authors of [29,30,31] all propose different techniques for self-localization using landmarks or beacons and triangulation concepts, while the authors of [25,26,32,33] all use trilateration.

In the literature mentioned above the solutions are based on the assumption that he landmark positions are known which is not the case in our setup. Extensive research has been done to solve mobile device localization given several known mobile base stations [34]. However, in this paper, we relax this assumption into a more general case where the landmark positions are assumed to be completely unknown. Instead we assume to know imprecise information about the sensor’s new position with respect to the previous one. This kind of information is actually the control commands to the robot which is always available for robot moving in its environment. Furthermore, to make the calibration process independent on the robot dynamical model, we assume to take measurements only when the robot is not moving (stand still).

Scan-matching techniques like ICP [35] and NDT [36,37] are also commonly used to localize a sensor in space. This class of methods determines the relative transformation between two lidar scans by minimizing surface-to-surface distances using all points in the scans, as opposed to a sparse set of extracted landmarks as is the case with triangulation and trilateration. Scan matching could potentially be used instead of, or in addition to, the control commands used as input to estimate the sensor pose in this work. However, this pose estimation is not the focus of the present paper, but rather the calibration of the range-dependent sensor noise.

Finally, we note that our strategy is specifically designed for cheap distance sensors: generally, the more accurate and precise a sensor is, the more useful (and, at the same time, likely expensive) it is. Our focus is on enabling software-based improvements of cheap sensors so that, by adding a bit of statistical processing, we extend their applicability. For this reason, we use triangulation lidar sensors as a practical and motivating example. Therefore, our paper relates also with the literature around the calibration of these sensors, and thus to the analyses on the effect of color of the target on the measured distance [38]; the works in [39,40], that build two partially different statistical models (the former homoskedastic, the latter heteroskedastic) and thus two slightly different calibration procedures based on ground truth information, using Weighted Least Squares (WLS) for parameter estimation and Akaike Information Criterion (AIC) for model selection; and the work in [41], that extends upon the work in [40] by including the effects of beam angles in the calibration process.

### 1.2. Statement of Contributions

Summarizing, we propose and validate a strategy that uses triangulateration concepts for calibrating distance sensors that return 2D measurements (i.e., both angles and distances). The algorithm leverages on placing the distance sensor inaccurately in equally distant positions along straight paths and making the measurements from the sensor comply with the assumption that the landmarks do not move, plus some other practical assumptions listed exhaustively in Section 2. The strategy is intended to be applicable at least (but not only) for the very specific situation where vacuum cleaning robots move within an apartment, and are able—by moving around and detecting obstacles—to self-calibrate their distance sensors.

More specifically, our strategy works as follows. First, we assume the pre-existence of a procedure that correctly identifies and distinguishes landmarks in the 2D measurements stream. Then, laddering on this knowledge, we perform two steps: (1) use the measured angles to the identified landmarks and the knowledge of the sensor movement to obtain an unbiased estimate of the landmarks positions in the fixed frame, and (2) calibrate the distance measurement model using these estimated landmarks positions.

We moreover validate the strategy using two approaches: (a) via simulated datasets, to analyze the limitations of the proposed procedure in a Monte Carlo (MC) fashion, and (b) to quantitatively assess the performance of the proposed procedure in real-life scenarios via field datasets recorded in a lab equipped with high-fidelity motion capture system.

### 1.3. Organization of the Manuscript

Section 2 formulates the calibration problem from a mathematical perspective. Section 3 describes the proposed algorithm, while Section 4 quantitatively assesses its performance. Section 5 concludes by listing the most important discoveries and research questions opened in the process.

## 2. Problem Formulation

We pose the following assumptions.

(A1)The environment from which we collect the measurements to be used for the calibration process has particular structures that produce easily recognizable features in the sensor readings. For example, the situation is as in Figure 3, where corners and poles produce clear features in the 2D plane of the measurements. Note that this means that our strategy cannot work in environments that miss these easily recognizable structures (such as natural places like deserts, or flat areas without trees). However, generally we consider applications where robots shall move precisely in the surroundings, and this calls for objects to be avoided. If there are no such obstacles/structures then the need for precise calibration becomes feeble. Given this, without loss of generality we require static and detectable landmarks; in this paper, we will use cylinders with known radius, but it could be anything as long as we have a detector for it.(A2)The sensor measurements lie in a 2D plane that is parallel to the ground. Moreover, the objects that produce the above mentioned features develop orthogonally w.r.t. the ground. This implies that the distances measurements are not affected by tilt effects. This requirement may not hold in generic situations; however, our envisioned calibration strategy is to be carried out within a building, where the conditions above hold. The problem of removing these assumptions is considered as a potential future extension.(A3)The statistical model underlying the distance readings contains heteroskedastic noise (for which the variance of the noise increases with the actual distance that shall be measured) and a bias whose amplitude also increases with the distance above. More specifically, we will focus on the situation where there exist l=1,…,L objects in the environment, and k=1,…,K places where the sensor can be placed. We then let $\left[x_{l},y_{l}\right]$ and $\left[{\tilde{x}}_{k},{\tilde{y}}_{k}\right]$ be, respectively, the Cartesian coordinates of the *L* objects and of the *K* sensor positions. Accordingly, the actual distance between the sensor position *k* and the object placement *l* is
(2)$$d_{l,k}=\sqrt{{\left(x_{l}-{\tilde{x}}_{k}\right)}^{2}+{\left(y_{l}-{\tilde{y}}_{k}\right)}^{2}}.$$We then assume that the distance readings are distributed as the polynomial model
(3)$${\tilde{d}}_{l,k}=\underset{\mathrm{bias}}{\underbrace{\sum_{i=0}^{n_{b}}\alpha_{i}{\left(d_{l,k}\right)}^{i}}}+\underset{\mathrm{heteroskedastic}\;\mathrm{noise}}{\underbrace{\sum_{i=0}^{n_{h}}\beta_{i}{\left(d_{l,k}\right)}^{i}e_{l,k}}}$$
with $e_{l,k}∼N\left(0,1\right)$ iid. The model parameters are thus $\left\{\alpha_{i}\right\}$, $\left\{\beta_{i}\right\}$, with *n* being the corresponding model order (hereafter assumed for simplicity equal for both the bias and noise terms, i.e., $n=n_{b}=n_{h}$). Note that in the following we may also use a simplified distance model that, for the sake of numerical tractability, neglects the heteroskedastic term in (Equation 3) so that the model reduces to
(4)$${\tilde{d}}_{l,k}=\sum_{i=0}^{n}\alpha_{i}{\left(d_{l,k}\right)}^{i}+e_{l,k}.$$We will refer to this model as to the “*simplified distance model*”.(A4)Finally, we also assume that the angular readings ${\tilde{\theta }}_{l,k}$ are noisy measurements of the actual angles $\theta_{l,k}$ from which the object *l* is seen by the sensor at position *k* with respect to the reference frame of the horizontal axis. More precisely, we assume
(5)$${\tilde{\theta }}_{l,k}=\theta_{l,k}+\nu_{l,k}$$
where the measurement noise is $\nu_{l,k}∼N\left(0,\sigma_{\theta }^{2}\right)$ iid. Note that in practice this is a simplificative assumption that we use for analytical tractability reasons and that, a posteriori, is motivated by the numerical results we got during our experiments (For the sake of precision, it would be more formally correct to model the angle measurement noise through a Von Mises distribution with circular mean and noncircular concentration parameter. However, such a distribution converges to a normal one as the concentration parameter grows larger. In our case thus the approximation is justified in practice). Note, moreover, that this assumption implies that $\sigma_{\theta }^{2}$ is an unknown parameter of the model. We also assume that the error characteristics of (Equation 3)–(Equation 5) are time-invariant and do not depend on the absolute positions of the landmark (while they obviously depend on the relative distances “sensor vs. obstacle”). The angle $\theta_{l,k}$ is thus the sum of the angle from which the object *l* is seen by the sensor with respect to the robot reference frame, plus the robot heading angle plus the rotation angle of the lidar’s internal coordinate system with the robot coordinate system which is assumed to be a known constant. Note also that the measurement noise $\nu_{l,k}$ in (Equation 5) incorporates imprecisions in the knowledge of the robot heading and rotation angle.

Summarizing, the calibration procedure shall return a reasonable model order $\hat{n}$ and an estimated parameter vector $\hat{\Theta }=\left[{\hat{\alpha }}_{1},\ldots ,{\hat{\alpha }}_{\hat{n}},{\hat{\beta }}_{1},\ldots ,{\hat{\beta }}_{\hat{n}},{\hat{\sigma }}_{\theta }^{2}\right]$. The problems are thus

(P1)design a statistically optimal or near-optimal (in the Mean Squared Error (MSE) sense) algorithm that can be computed using closed-form expressions, and that can simultaneously estimate: the sensor coordinates $x_{k},y_{k}$ for each sampling position *k*, the position of the objects $x_{l},y_{l}$ for each object *l*, the model order $\hat{n}$ and the model parameters vector $\hat{\Theta }$ above;(P2)quantitatively characterize the statistical performance of these estimators using appropriate mathematical analysis and field tests.

## 3. A Triangulateration Strategy for Calibrating Distance Sensors

Optimally solving the problem 2 above requires jointly solving a nonlinear parameter estimation and a nonlinear state estimation problem. The solution is in general not available in closed form, and a viable numerical strategy could be using Monte Carlo techniques. However, this would require extensively long simulations and high processing power, which we assume is not available or usable. Recall indeed our initial idea: our goal is to develop strategies that can be used to endow cheap embedded systems (such as vacuum cleaning robots) to autonomously self-calibrate their distance sensors when desired and without the need to connect to external computing infrastructure. Therefore, we proceed to solve the problem using an ad hoc strategy that is easily implementable on normal embedded systems at the cost of sacrificing optimality of the estimates in the MSE sense.

More precisely, we propose to construct an estimator that computes solutions performing the following steps:Assume to know that there exist *L* landmarks, and to be able to identify and label them at each time instant from the raw measurements stream;place the sensor in a finite number of ideally equally spaced positions along an ideally straight line (say $s_{k}$ where k=1,…,K);collect noisy measurements of the angles ${\tilde{\theta }}_{l,k}$ and distances ${\tilde{d}}_{l,k}$ between the sensor and the various landmarks l=1,…,L at each position $s_{k}$ (we recall that the stochastic models for these processes are (Equation 3) and (Equation 5));estimate the 2D positions of the *L* landmarks in the inertial frame based on the sensor angle measurements ${\tilde{\theta }}_{l,k}$ only, using the strategy defined in Section 3.1 below; andgiven the estimated landmark positions above, and the measured distances ${\tilde{d}}_{l,k}$, estimate the model order and model parameters (i.e., do the actual sensor calibration step) with the strategy proposed in Section 3.2 below.

### 3.1. Estimating the 2D Positions of Circular Landmarks

For illustration purposes and to make the paper self-contained, we now present a simple landmark position estimation algorithm and show that the method still works, even though better landmark detectors may be available. We also note that the focus of the paper is on sensor calibration. However, the same method developed for the sensor calibration part can be used to build a simple landmark position estimation algorithm. Recall then that we assume the landmarks l=1,…,L to be circular geometrical features in the measurement stream that are induced by distinct circular objects in the sensor environment as in Figure 2. To contextualize this assumption, consider Figure 3 and Figure 4 and their captions, showing the sensor mounted on a robot moving in between the various circular landmarks. We used a lidar sensor from Neato. (It measure ranges from 0.2 m to 6 m and cover full 360 degree planar scan with angular resolution of 1 degree, [42]). (Neato Robotics www.neatorobotics.com).

Assume then, as in step 2 above, to place the sensor in k=1,…,K ideally equally spaced positions along an ideally straight line, and to collect the corresponding raw measurements from the sensor. Note that it is possible to do the calibration without moving in a straight line, but it was chosen so to prevent the error from increasing largely during rotation and also to keep motion model (Equation 8) as simple as possible by excluding the robot heading angle. We selected equally spaced positions to minimize the error propagation from rover controller to the calibration process, as different step sizes might have different error levels in the controller. The next step is to compute, starting from this raw data, an estimate of the center of each circular landmark *l* using the information obtained at each sensor position *k*. Intuitively, we estimate the center of the landmark *l* as the point that minimizes the sum of its distances with the lines obtained at each *k* pointing towards the landmark, as shown in Figure 5 and described more formally in its caption.

Given that in this paper we assume that the sensor sees round landmarks, this means that we implicitly assume the presence of an offset in the distance measurements that is equal to the landmark radius. For brevity, here we assume to know this parameter. Otherwise, circular landmarks lead to raw measurements like the ones in Figure 2, from which it is not difficult to obtain practically accurate estimates of such radii.

Therefore, to summarize, we assume that the robot moves along a straight line and that the sensor takes measurements in equally spaced positions along this line. For each landmark we can compute the straight lines that aim from all the various sensor positions to the (individually estimated) landmark centers. If there was no error, we could find the center of each landmark simply by intersecting the various lines referring to the same landmark. As we are in practice always far from this ideal condition (i.e., do not have accurate information neither about the sensor positions nor the measurement angles), the proposal is to find an estimate of each landmark center, say ${\hat{x}}_{l},{\hat{y}}_{l}$, with a least-squares solution that minimizes the sum of perpendicular distances from the unique solution point to all these lines.

The remainder of this section is then dedicated to deriving the analytical structure of such estimator. To help readability, the notation is general so that *l* means “landmark”, $s_{k}$ means “sensor position”, *x* and *y* relate to the Cartesian reference frame, ★ and $\tilde{★}$ refer to respectively the “ideal” and “noisy” versions of the same quantity. For example, the relation
(6)$${\tilde{\theta }}_{l,k}=\theta_{l,k}+\nu_{l,k}$$
indicates that the measured angle ${\tilde{\theta }}_{l,k}$ of landmark *l* w.r.t. the position $s_{k}$ is a noisy version of the actual angle $\theta_{l,k}$.

To find the estimated landmark position ${\hat{x}}_{l},{\hat{y}}_{l}$ in a closed form, consider then that in its *k*-th position the sensor is located at $s_{k}={\left[{\tilde{x}}_{k}\;{\tilde{y}}_{k}\right]}^{T}$ (importantly, a quantity that is unknown to the system). Ideally, the (k+1)-th position of the sensor should be along a straight line (i.e., a fixed heading angle) and at a fixed distance, i.e., be
(7)$${\left[x_{k+1}\;y_{k+1}\right]}^{T}={\left[x_{k}\;y_{k}\right]}^{T}+{\left[\delta x_{k}\;\delta y_{k}\right]}^{T}$$
where $\delta x_{k}$ and $\delta y_{k}$ are either zeros when the robot is not moving or constants determined by the step size when the robot is moving. In practice, though, the ideal conditions are not satisfied. We thus model the actual sensor position to be
(8)$${\left[{\tilde{x}}_{k+1}\;{\tilde{y}}_{k+1}\right]}^{T}={\left[x_{k+1}\;y_{k+1}\right]}^{T}+{\left[e_{x,k}\;e_{y,k}\right]}^{T}$$
where $e_{x,k}$ and $e_{y,k}$ are zero mean Gaussian iid with the same variance $\sigma_{s}^{2}$. Note that this Gaussianity assumption is once again instrumental for the purpose of being able to devise computationally efficient schemes that can be implemented in embedded systems. Note, moreover, that we are recording the sensor measurements after the robot reached its new position (i.e., we are not considering measurements recorded during transients). As the actual sensor positions $s_{k}$ are not available, the best we have is only the reference (noiseless) sensor positions ${\left[x_{k}\;y_{k}\right]}^{T}$ which can be determined using (Equation 7).

Moreover, consider the actual sensor position $s_{k}$ and the measured angle ${\tilde{\theta }}_{l,k}$ of landmark *l* w.r.t. the position $s_{k}$ (line whose slope is then $\tan({\tilde{\theta }}_{l,k})$, as the dotted lines in Figure 5). The equation of this line is then
(9)$$\sin\left({\tilde{\theta }}_{l,k}\right)x_{l}-\cos\left({\tilde{\theta }}_{l,k}\right)y_{l}-\sin\left({\tilde{\theta }}_{l,k}\right){\tilde{x}}_{k}+\cos\left({\tilde{\theta }}_{l,k}\right){\tilde{y}}_{k}=0.$$

Substituting (Equation 5) in (Equation 9) above yields
(10)$$\begin{array}{cc} \; & \sin\left(\theta_{l,k}+\nu_{l,k}\right)x_{l}-\cos\left(\theta_{l,k}+\nu_{l,k}\right)y_{l}-\\ \; & \sin\left(\theta_{l,k}+\nu_{l,k}\right){\tilde{x}}_{k}+\cos\left(\theta_{l,k}+\nu_{l,k}\right){\tilde{y}}_{k}=0. \end{array}$$

Expanding then the sine and cosine terms using the trigonometric identities
(11)$$\begin{array}{cc} \sin(\theta_{l,k}+\nu_{l,k}) & =\sin\left(\theta_{l,k}\right)\cos\left(\nu_{l,k}\right)+\cos\left(\theta_{l,k}\right)\sin\left(\nu_{l,k}\right) \end{array}$$
(12)$$\begin{array}{cc} \cos(\theta_{l,k}+\nu_{l,k}) & =\cos\left(\theta_{l,k}\right)\cos\left(\nu_{l,k}\right)-\sin\left(\theta_{l,k}\right)\sin\left(\nu_{l,k}\right) \end{array}$$
and simplifying gives the following expanded equation
(13)$$\begin{array}{cc} \; & \left(\sin\left(\theta_{l,k}\right)+\cos\left(\theta_{l,k}\right)\tan\left(\nu_{l,k}\right)\right)x_{l}-\\ \; & \left(\cos\left(\theta_{l,k}\right)-\sin\left(\theta_{l,k}\right)\tan\left(\nu_{l,k}\right)\right)y_{l}-\\ \; & \left(\sin\left(\theta_{l,k}\right)+\cos\left(\theta_{l,k}\right)\tan\left(\nu_{l,k}\right)\right){\tilde{x}}_{k}+\\ \; & \left(\cos\left(\theta_{l,k}\right)-\sin\left(\theta_{l,k}\right)\tan\left(\nu_{l,k}\right)\right){\tilde{y}}_{k}=0. \end{array}$$

Moving the stochastic terms to the right hand side of the equation, we then obtain
(14)$$\begin{array}{cc} \; & \sin\left(\theta_{l,k}\right)x_{l}-\cos\left(\theta_{l,k}\right)y_{l}-\sin\left(\theta_{l,k}\right){\tilde{x}}_{k}+\cos\left(\theta_{l,k}\right){\tilde{y}}_{k}\\ \; & =g_{k}\tan\left(-\nu_{l,k}\right) \end{array}$$
where
(15)$$g_{k}:=\sin\left(\theta_{l,k}\right)\left(y_{l}-{\tilde{y}}_{k}\right)+\cos\left(\theta_{l,k}\right)\left(x_{l}-{\tilde{x}}_{k}\right).$$

Now substituting the geometrical identities $\sin\left(\theta_{l,k}\right)=\frac{\left(y_{l}-{\tilde{y}}_{k}\right)}{d_{l,k}}$ and $\cos\left(\theta_{l,k}\right)=\frac{\left(x_{l}-{\tilde{x}}_{k}\right)}{d_{l,k}}$ (immediately proved upon inspecting Figure 6) into (Equation 15), and then simplifying leads to
$$g_{k}:=\frac{{\left(y_{l}-{\tilde{y}}_{k}\right)}^{2}+{\left(x_{l}-{\tilde{x}}_{k}\right)}^{2}}{d_{l,k}}=d_{l,k}.$$

Recall that $d_{l,k}$ is the ground truth for the sensor-to-landmark distances—a ground truth that is not available. Consider then that our main goal is to calibrate the sensor without using such ground truth information. Therefore, the best we can do instead is to plug in the measured distances ${\tilde{d}}_{l,k}$ as estimates (or the sample mean if more than one measurement is available at the same sensor position).

For small values of $\nu_{l,k}$ (i.e., for the case where the dotted lines in Figure 5 aim decently at their target) we can simplify (Equation 14) using the approximations
(16)$$\sin\left(\nu_{l,k}\right)≊\nu_{l,k}\;\cos\left(\nu_{l,k}\right)≊1.$$

This means obtaining
(17)$$\begin{array}{cc} \; & \sin\left(\theta_{l,k}\right)x_{l}-\cos\left(\theta_{l,k}\right)y_{l}-\sin\left(\theta_{l,k}\right){\tilde{x}}_{k}+\cos\left(\theta_{l,k}\right){\tilde{y}}_{k}\\ \; & ≊{\tilde{d}}_{l,k}\nu_{l,k}. \end{array}$$

Recall then that in our assumptions the measurement noise $e_{l,k}$ in (Equation 3) is statistically independent of the measurement noise $\nu_{l,k}$ in (Equation 5). For that reason, the residual error in (Equation 17) will be heteroskedastic since the variance is $\sigma_{\theta }^{2}$ multiplied by the heteroskedastic variance of ${\tilde{d}}_{l,k}$ (that, we recall, is different in each sensor position). In the ideal case of no measurement noise, the line should pass through the center of the landmark *l*. In this special case, the point of intersection between these lines will be the position of the landmark center. As mentioned above, the presence of the measurement noise will however make the lines drift. The *K* lines corresponding to the *K* sensor positions in general will not intersect in a unique point, but in pairs. In this case, we may then solve the problem in a least-squares sense: the idea is to minimize the weighted sum of the squared distances (the solid red lines in Figure 5), i.e., to solve
(18)$$\left[\begin{array}{c} {\hat{x}}_{l}\\ {\hat{y}}_{l} \end{array}\right]=\arg\min_{x_{l},y_{l}\in R}\;{∥W_{l}^{\frac{1}{2}}\left(H_{l}\left[\begin{array}{c} x_{l}\\ y_{l} \end{array}\right]-{\tilde{b}}_{l}\right)∥}^{2}$$
where
(19)$$\begin{array}{cc} W_{l} & :=\left[\begin{array}{ccccccc} \frac{1}{\mathrm{var}\left({\tilde{d}}_{l,1}\right)} & \; & 0 & \; & \cdots & \; & 0\\ 0 & \; & \frac{1}{\mathrm{var}\left({\tilde{d}}_{l,1}\right)} & \; & \cdots & \; & 0\\ \vdots & \; & \vdots & \; & \ddots & \; & \vdots \\ 0 & \; & \cdots & \; & 0 & \; & \frac{1}{\mathrm{var}\left({\tilde{d}}_{l,1}\right)} \end{array}\right]\\ H_{l} & :=\left[\begin{array}{ccc} \sin(\theta_{l,1}) & \; & -\cos(\theta_{l,1})\\ \vdots & \; & \vdots \\ \sin(\theta_{l,K}) & \; & -\cos(\theta_{l,K}) \end{array}\right]\\ {\tilde{b}}_{l} & :=\left[\begin{array}{c} \sin\left(\theta_{l,1}\right){\tilde{x}}_{1}-\cos\left(\theta_{l,1}\right){\tilde{y}}_{1}\\ \vdots \\ \sin\left(\theta_{l,K}\right){\tilde{x}}_{K}-\cos\left(\theta_{l,K}\right){\tilde{y}}_{K} \end{array}\right]. \end{array}$$

Consequently, solving this system according to Aitken’s generalized least square method [43] gives the following Best Linear Unbiased Estimator (BLUE) of the landmark centers,
(20)$$\left[\begin{array}{c} {\hat{x}}_{l}\\ {\hat{y}}_{l} \end{array}\right]={\left(H_{l}^{T}W_{l}H_{l}\right)}^{-1}W_{l}H_{l}^{T}{\tilde{b}}_{l}.$$

The computations above assume the full knowledge of the sensor positions $s_{k}=\left[{\tilde{x}}_{k}\;{\tilde{y}}_{k}\right]$. As this information is not available, the best we can do is to plug in, instead, the expected sensor positions $\left[x_{k}\;y_{k}\right]$ in (Equation 7). Replacing $\left[{\tilde{x}}_{k}\;{\tilde{y}}_{k}\right]$ with $\left[x_{k}\;y_{k}\right]$ in the least squares problem (Equation 18) thus leads to the problem
(21)$$\left[\begin{array}{c} {\hat{x}}_{l}\\ {\hat{y}}_{l} \end{array}\right]=\arg\min_{x_{l},y_{l}\in R}\;{∥{W_{l}^{*}}^{\frac{1}{2}}\left(H_{l}\left[\begin{array}{c} x_{l}\\ y_{l} \end{array}\right]-b_{l}\right)∥}^{2}$$
where
(22)$$\begin{array}{cc} W_{l}^{*} & :=W_{l}\;\left(\mathrm{as}\mathrm{explaind}\mathrm{below}\mathrm{Equation}\;\left(25\right)\right)\\ b_{l} & :=\left[\begin{array}{c} \cos\left(\theta_{l,1}\right)y_{1}-\sin\left(\theta_{l,1}\right)x_{1}\\ \vdots \\ \cos\left(\theta_{l,K}\right)y_{K}-\sin\left(\theta_{l,K}\right)x_{K} \end{array}\right] \end{array}$$
which in turns gives the weighted least squares estimator
(23)$$\left[\begin{array}{c} {\hat{x}}_{l}\\ {\hat{y}}_{l} \end{array}\right]={\left(H_{l}^{T}W_{l}^{*}H_{l}\right)}^{-1}W_{l}^{*}H_{l}^{T}b_{l},$$
whose weights matrix $W^{*}$ is motivated below and defined in (Equation 25).

This estimator is *solvable*, in the sense that the embedded system has all the information necessary to compute and minimize the cost. In other words, solving (Equation 21) is computationally feasible as all the required information is available while solving (Equation 18) is not. However, solving (Equation 21) leads to solving an approximate model of the intersection problem which will indeed result in a biased estimator, as it corresponds to solving the system of equations that is obtained after substituting $x_{k}$ and $y_{k}$ from (Equation 8) into (Equation 17), i.e.,
(24)$$\begin{array}{cc} \; & \sin\left(\theta_{l,k}\right)x_{l}-\cos\left(\theta_{l,k}\right)y_{l}-\sin\left(\theta_{l,k}\right)x_{k}+\cos\left(\theta_{l,k}\right)y_{k}\\ \; & ≊\sin\left(\theta_{l,k}\right)e_{x,k}-\cos\left(\theta_{l,k}\right)e_{y,k}+{\tilde{d}}_{l,k}\nu_{l,k}. \end{array}$$

To characterize the error of this estimator, we notice that the residual error includes two different terms: the first is homoskedastic (specifically, corresponding to the first two terms in (Equation 24)) and with a variance of
$$\begin{array}{cc} \mathrm{var}\left(\sin\left(\theta_{l,k}\right)e_{x,k}-\cos\left(\theta_{l,k}\right)e_{y,k}\right) & =\sin{\left(\theta_{l,k}\right)}^{2}\sigma_{s}^{2}+\cos\left(\theta_{l,k}^{2}\right)\sigma_{s}^{2}\\ & =\sigma_{s}^{2}. \end{array}$$

The second term is heteroskedastic with variance $\sigma_{\theta }^{2}\;\mathrm{var}\left({\tilde{d}}_{l,k}\right)$ (specifically, corresponding to the last terms in (Equation 24)). Consequently, we suggest to set the weighting matrix $W_{l}^{*}$ as
(25)$$W_{l}^{*}:=\sigma_{s}^{-2}I+\sigma_{\theta }^{-2}W_{l}.$$

Note that in our assumptions both $\sigma_{s}^{2}$ and $\sigma_{\theta }^{2}$ are assumed unknown constants. However, minimizing the sum of the squared residual errors in (Equation 24) is equivalent to minimizing the sum of the squared residual errors ${\tilde{d}}_{l,k}\nu_{l,k}$, as the transformation between these errors is affine. This means that replacing $W_{l}^{*}$ with $W_{l}$ will give exactly the BLUE for the parameters in (Equation 24).

### 3.2. Calibrating the Sensor

Once all the *L* landmark positions are estimated as ${\hat{x}}_{l},{\hat{y}}_{l}$ as in the previous section, we can easily estimate the various landmark–sensor position distances ${\hat{d}}_{l,k}$ simply through computing the distance between each landmark with all the sensor positions and subtracting the radius of the landmark (which, as we said above, is either assumed to be known or assumed to be inferrable from the raw data). For notational compactness, define the (KL×1)-dimensional distance measurement vector
$$\tilde{d}:={\left[\begin{array}{ccccccc} {\tilde{d}}_{1,1} & \cdots & {\tilde{d}}_{1,K} & \cdots & {\tilde{d}}_{L,1} & \cdots & {\tilde{d}}_{L,K} \end{array}\right]}^{T},$$
the noise vector
$$e:={\left[\begin{array}{ccccccc} e_{1,1} & \cdots & e_{1,K} & \cdots & e_{L,1} & \cdots & e_{L,K} \end{array}\right]}^{T},$$
and rewriting (Equation 4) through a Vandermonde matrix, i.e.,
(26)$$\tilde{d}=\underset{:=\Phi }{\underbrace{\left[\begin{array}{ccccccccc} 1 & \; & {\hat{d}}_{1,1} & \; & {\left({\hat{d}}_{1,1}\right)}^{2} & \; & \cdots & \; & {\left({\hat{d}}_{1,1}\right)}^{n}\\ \vdots & \; & \vdots & \; & \vdots & \; & \ddots & \; & \vdots \\ 1 & \; & {\hat{d}}_{1,K} & \; & {\left({\hat{d}}_{1,K}\right)}^{2} & \; & \cdots & \; & {\left({\hat{d}}_{1,K}\right)}^{n}\\ \vdots & \; & \vdots & \; & \vdots & \; & \ddots & \; & \vdots \\ 1 & \; & {\hat{d}}_{L,1} & \; & {\left({\hat{d}}_{L,1}\right)}^{2} & \; & \cdots & \; & {\left({\hat{d}}_{L,1}\right)}^{n}\\ \vdots & \; & \vdots & \; & \vdots & \; & \ddots & \; & \vdots \\ 1 & \; & {\hat{d}}_{L,K} & \; & {\left({\hat{d}}_{L,K}\right)}^{2} & \; & \cdots & \; & {\left({\hat{d}}_{L,K}\right)}^{n} \end{array}\right]}}\underset{:=\alpha }{\underbrace{\left[\begin{array}{c} \alpha_{0}\\ \alpha_{1}\\ \alpha_{2}\\ \vdots \\ \alpha_{n} \end{array}\right]}}+e$$
where the Vandermonde matrix Φ is of size KL×n+1 and the parameter vector of size n+1×1.

Given this notation, the calibration procedure consists of three phases:*phase#1: model parameters estimation*. After obtaining the estimates of the distances between the sensor and landmarks, estimate the parameters α casting the problem as a linear regression on (Equation 26) and the measurement vector $\tilde{d}$ for model orders $n=0,1,2,\ldots ,n_{\max}$, where $n_{\max}$ is a user-defined parameter. This means solving for each potential *n* the problem
(27)$$\hat{\alpha }=\arg\min_{\alpha \in R^{n+1}}\;{∥\Phi \alpha -\tilde{d}∥}^{2}$$
which has the closed-form solution
(28)$$\hat{\alpha }={\left(\Phi^{T}\Phi \right)}^{-1}\Phi^{T}\tilde{d}.$$Note that, once again, the estimator $\hat{\alpha }$ is unbiased; however, due to the simplification of the noise term in (Equation 3) (i.e., ignoring the heteroskedastic part of the noise), $\hat{\alpha }$ will not be efficient.*phase#2: model order selection*. We note that there exist various alternatives for selecting the optimal model order $\hat{n}\in \left\{0,1,2,\ldots ,n_{\max}\right\}$: fitting opportune test sets, using crossvalidation, or also using model order selection criteria, for example, AIC. In the setups we considered for this paper we actually found that the model order selection problem has quite clear solutions, implying that all the various alternatives clearly indicated the very same number (see Section 4), implying in its turn that for our specific case all the various approaches tend to give equivalent results. It may, however, be that in other cases different strategies lead to different results;*phase#3: filtering new measurements*. Once the model order selection and the model parameters estimation problems are solved, this means rewriting the “object distance vs. sensor reading” measurement model (Equation 3) as
(29)$$\tilde{d}=\sum_{i=0}^{\hat{n}}{\hat{\alpha }}_{i}{\left(d\right)}^{i}+\sum_{i=0}^{\hat{n}}{\hat{\beta }}_{i}{\left(d\right)}^{i}e$$
where $\tilde{d}$ is the raw measurement, and *d* is the actual distance. To estimate *d* from $\tilde{d}$ and the trained model one should thus invert (Equation 29). This inversion is not immediate; for example, one may solve the Least Squares (LS)-type optimization problem
(30)$$\hat{d}=\arg\min_{\hat{d}\in R_{+}}\;{∥\sum_{i=0}^{\hat{n}}{\hat{\alpha }}_{i}{\hat{d}}^{i}-\tilde{d}∥}^{2}$$
which requires finding the roots of a polynomial of order $2\hat{n}$. Thus, despite its apparent simplicity, the problem of finding polynomial roots requires numerical methods for polynomial orders greater than 3.

## 4. Numerical Results

In this section, we verify in an empirical way the performances of the proposed calibration procedure, first with simulations using Matlab® and then with laboratory experiments using real sensors and landmarks in an environment endowed with a localization infrastructure that can return accurate assumingly ground truth information.

In general terms we thus consider ${\tilde{d}}_{j}$, j=1,…,J raw measurements from a noncalibrated sensor. To each raw distance measurement ${\tilde{d}}_{j}$ there corresponds also a true distance $d_{j}$, and a filtered distance ${\hat{d}}_{j}$, i.e., the corresponding filtered version of these raw data.

As for the statistical performance index, the goal is to assess if and how much the calibration algorithm is actually leading to improved estimates of the distances, i.e., whether ${\hat{d}}_{j}$ is statistically closer to $d_{j}$ than ${\tilde{d}}_{j}$, and if so how much. To do so we use the **MSE**, i.e.,
$$\mathrm{MSE}\left(a,b\right):=\frac{1}{J}\sum_{j=1}^{J}{\left(a_{j}-b_{j}\right)}^{2}.$$

More precisely we will compute the ratio between the **MSE** computed with the raw data (distances measured by the sensor) and the **MSE** computed with the estimated data (distances estimated by the algorithm), i.e., use the
(31)$$\mathrm{MSE}\mathrm{ratio}:=\frac{\mathrm{MSE}(d,\tilde{d})}{\mathrm{MSE}(d,\hat{d})}.$$

Thus, in order to get an improvement with the estimation, this ratio has to be greater than 1.

### 4.1. Analyzing the Statistical Properties of the Landmark Position Estimator through Simulation Results

Unless otherwise stated, for all our simulations’ plots, each point is the average of 1000 simulations of five landmarks and 20 sensor positions.

We start with statistically characterizing the landmark position estimator described in Section 3.1 by simulating a measurement model of the type (Equation 3) characterized by n=2, α=[0.0525,0.8838,0.0584] and β=0.05α, values that seem representing typical distance measurement systems mounted in modern autonomous vacuum systems.

We then investigate the MSE of the landmark position estimation procedure by analyzing how its bias and variance depend on three specific quantities:the standard deviation $\sigma_{\theta }$ associated to the uncertainty of the sensor-to-landmark angle measurement in (Equation 6),the standard deviation $\sigma_{s}$ associated to the uncertainty in the sensor position evolution in (Equation 8), andthe total number of landmarks *L* present in the scene.

The results are summarized in Figure 7, plotting the dependencies on $\sigma_{s}$ for a set of given $\sigma_{\theta }$ and *L*, and in Figure 8, plotting the dependencies on *L* given $\sigma_{s}$ and $\sigma_{\theta }$.

In words, the results shown in Figure 7 and Figure 8 confirm the obvious intuition that the smaller the noises, the better the estimator. However, we also note that, from numerical standpoints, it seems that guaranteeing $\sigma_{s}<10$ cm is important, and that guaranteeing $\sigma_{s}<1$ cm is instead not a necessity. This result is of practical importance, because the assumption that the sensor will be placed in a perfectly straight line will always be violated. However it seems that, at least for standard “domestic” cases like autonomous vacuum cleaners, violating this assumption will not disrupt the final results. We also note that Figure 8 suggests to set *L* to be around 5, i.e., an environment that is sufficiently rich while not being cluttered.

### 4.2. Analyzing the Statistical Properties of the Sensor Calibration Procedure through Simulation Results

We then pass to the second part of the estimation procedure, i.e., calibrating the model parameter that we presented in Section 3.2. Recall that the calibration algorithm is based on the estimated landmarks position, i.e., there is the need to estimate the landmarks positions first, to then proceed to the calibration step. We then define as main performance index the MSE ratio (Equation 31), i.e., a measure of how much worse the raw data measured by the sensor is w.r.t. the distances estimated by the data filtering algorithm. We analyze how the MSE ratio (Equation 31) depends on the standard deviation $\sigma_{\theta }$ of the sensor-to-landmark angle measurement error in (Equation 6), and the standard deviation $\sigma_{s}$ of the sensor position evolution uncertainty in (Equation 8).

The results are summarized in Figure 9, and they show that the overall approach seems to be robust: increasing gradually the standard deviations $\sigma_{\theta }$ and $\sigma_{s}$ does not lead to abrupt decays of the overall statistical performance. Moreover, for values of $\sigma_{\theta }$ and $\sigma_{s}$ that are meaningful in autonomous vacuum cleaners situations, we note MSE ratios that may reach 100.

We also remark that the overall strategy seems robust in its model order selection step. More precisely, in all our simulations we selected a model order n=2, which is the value we obtained in our previous work [40] while calibrating the same lidar sensor from field data (a value that is numerically convenient also because n=2 leads to closed-form solutions for (Equation 30)). In the simulations considered in this section, the overall estimation approach leads to estimating the model order $\hat{n}$ as the correct one, i.e., 2, when the standard deviations $\sigma_{\theta }$ and $\sigma_{s}$ are reasonably low. However, as the noises increase, we noted that the order selection process tends to become more and more conservative, and select the simpler model $\hat{n}=1$ (see graphically Figure 10).

Finally, we again investigate the effect of using different numbers of landmarks on the whole calibration process. The results, summarized in Figure 11, show again that increasing the number of landmarks from one to three leads to noticeable improvements in the MSE ratio. However, increasing the number of landmarks further will not lead to further improvements while, at the same time, increasing the computation complexity.

### 4.3. Field Experiments

We consider field experiments in a laboratory provided with a Vicon motion capture system that uses triangulation to compute the position of the objects inside the laboratory. Such a Vicon system is very accurate, compared to the sensors we aim at calibrating. For this reason, we assume that the Vicon measurements are for all the practical purposes noiseless and considerable as ground truth.

We then apply both the landmarks position estimator in Section 3.1 and the consequent parameters calibration procedure in Section 3.2 to calibrate the triangulation lidar shown in Figure 4. We recall that this type of lidar is not really accurate, as its measurements are affected by both a systematic bias and a heteroskedastic variance (see Figure 1) that lead to increasing measurement errors as the measured distance increases.

For practical purposes we placed the lidar sensor on top of a Pioneer 3AT mobile robot, as shown in Figure 4, controlled through a computer using Robot Operating System (ROS). We also consider five hand-made cylindrical landmarks with a radius of 12 cm, scattered within the field of view of the Vicon system and in a way that all of them are always visible and distinguishable by the sensor from all the positions $\left[x_{k},y_{k}\right]$ from where it will take measurements. As a practical indication, because of the intrinsic limits of the considered sensor, each landmark has to be placed not farther than 5 meters and closer than 20 cm from all the sensor positions. We then programmed the robot to move on a straight line path, and oriented it so to not hit the landmarks while moving. More precisely, we programmed the robot to move and stop 10 times, doing each time an incremental step of 30 cm. In order to have proper calibration of the sensor we need to have a “sufficient” calibration dataset. In general, for range sensors, a sufficient dataset should cover all the sensor ranges of interest. During our field experiments we moved the sensor in a straight line for about 3 m to ensure the richness of the recorded dataset.

Figure 3 shows a photo of one of the experiments. We repeated this type of experiment with three different placements, so to take three datasets of Vicon measurements of distances and angles (i.e., ground truth) of the five landmarks for all the sensor positions. In other words, thanks to the Vicon system we were able to compute all the actual distances and angles between all the various landmarks and the sensor in its various positions.

We then split each dataset in three parts: the first two to be used as a training set (the first third to estimate the sensor parameters in (Equation 3) and the second third to choose the model order $\hat{n}$) and the third part to be used as a test set. The field results presented for landmarks number other than 5 is basically obtained with the same recorded dataset of five landmarks after removing the receded data of extra land marks. For example, the 3-landmark datasets, is the 5-landmark datasets after removing the recorded data associated with the last two landmarks, and the 4-landmark datasets is the same as the 5-landmark datasets after removing the data of the fifth landmark, and so on. The obtained results, summarized in Figure 12, show again that the improvements change as more landmarks are involved in the calibration process. In other words, the field results are in good agreement with the simulated ones. However, a few outliers still exist which might be due to increased noise variance in one of the unconsidered processes like the landmark association problem. Moreover, in all the calibrations we performed on the different datasets, we obtain a selected model order $\hat{n}=1$, which indicates, based on our simulations, that there may be a high noise variance associated to the noise in measuring the angle with the landmarks.

Finally, Figure 13 shows how the proposed calibration procedure can help in real-life situations by reporting the measurement process relative to the third placement considered in Figure 12. Here, the landmarks’ borders are plotted as gray circles, the series of actual positions of the sensor as a stripe of blue dots. Moreover, the red crosses plot the raw measurements ${\tilde{d}}_{l,k}$ obtained by the sensor when observing the various landmarks, while the green dots plot the filtered measurements ${\hat{d}}_{l,k}$ obtained by applying the proposed calibration and filtering algorithms. One may note how the ${\hat{d}}_{l,k}$’s capture the actual positions of the various landmarks in a qualitatively much more precise way.

## 5. Conclusions

The nonlinear and heteroskedastic model of distance sensors can be calibrated by exploiting just the structure of a fixed environment. In other words, if the environment presents some particular features that may be used as generic landmarks, then one may use the fact that the landmarks do not move to infer the own movement. This means the possibility of estimating the landmarks’ positions minimizing opportune cost functions, and in this way obtain information useful to learn the characteristics of a distance sensor without the need for external distance measuring devices to be used as providers of ground truth information.

Through field experiments we saw that the overall proposed calibration approach may be quite robust: even if one does not get results that are as good as the ones achievable using external ground truth systems, our algorithm has been able to lead to a reduction of the norm of the measurement errors between the precalibration raw data and the postcalibration ones by a factor 10; in comparison, using ground truth calibration as in [40] led to a reduction factor of 17 (thus better but not of orders of magnitude, and at the cost of having to buy, set up and use a ground truth collection system).

We thus remark that this factor 10 achievement is through using just software logic and assumptions on the landmarks being fixed, and no additional hardware nor special conditions. In conclusion, the here proposed calibration procedure is expected to lessen the time to prepare the calibration setup, and is expected to be implementable well beyond laboratory setups.

We though recall that one standing assumption we exploited is that sensor measurements lie in a 2D plane that is parallel to the ground. As this requirement may not hold in some practical situations, we devise this as the most important future research direction spanned by the current work.

## Figures and Tables

**Figure 1 sensors-21-00155-f001:**
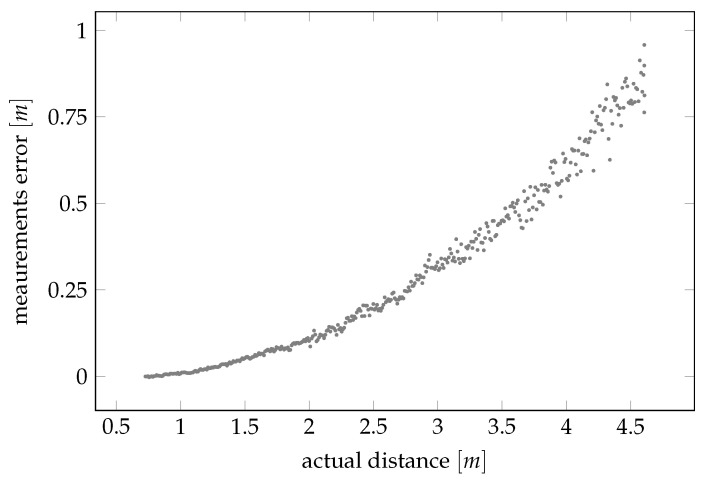
An example of a series of raw measurements obtained using a noncalibrated distance sensor (in this case the triangulation lidar described in Section 4.3).

**Figure 2 sensors-21-00155-f002:**
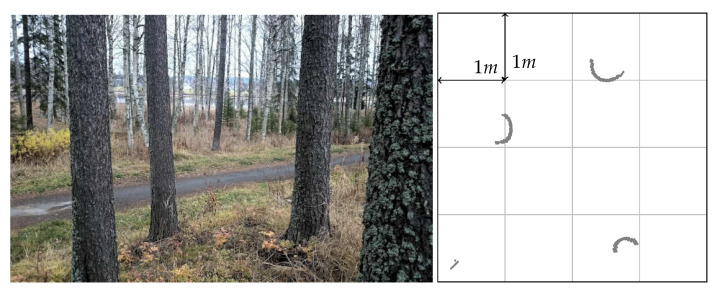
Raw measurements (right plot) from a noncalibrated triangulation lidar surrounded by trees in a forest (the ones in the left picture).

**Figure 3 sensors-21-00155-f003:**
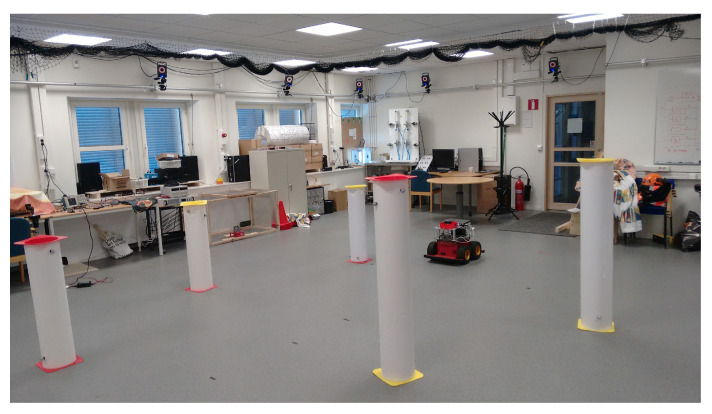
Setup of a typical calibration experiment, comprising five landmarks (white cylinders) and the robot–sensor system of Figure 4 moving among the landmarks.

**Figure 4 sensors-21-00155-f004:**
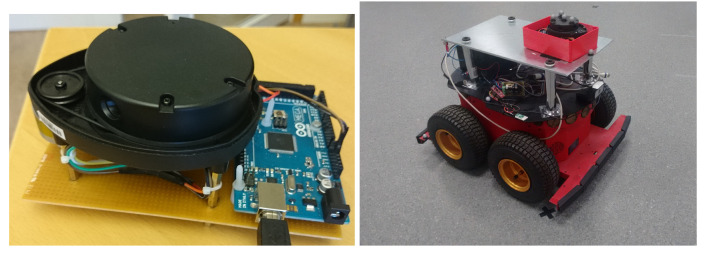
Photo of a mobile robot with a triangulation lidar mounted on its top.

**Figure 5 sensors-21-00155-f005:**
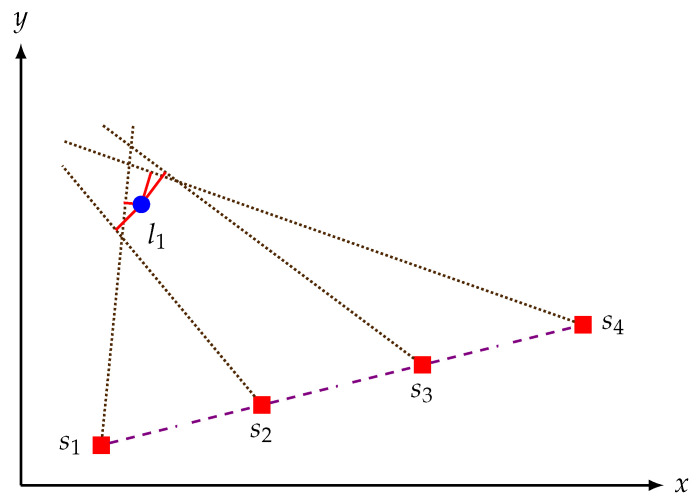
Illustration of the intuitions behind the suggested landmark position estimation algorithm. For each sensor position *k* one may identify from the raw distance measurements the angle between the sensor and the landmark, and thus the direction of the line from the sensor to the center of the landmark estimated while staying in position *k* (the dotted lines). Note that as while staying in position *k* the estimate of the center of the landmark is uncertain, these dotted lines do not aim perfectly at the actual center of the landmark, i.e., the blue dot labeled with $l_{1}$. An intuitively meaningful strategy for estimating the unknown position of $l_{1}$ is then finding that point that minimizes the sum of its distances with the dotted lines from each position *k*. Given this intuition, the short solid red lines represent the distances between these directions and the center of the landmark.

**Figure 6 sensors-21-00155-f006:**
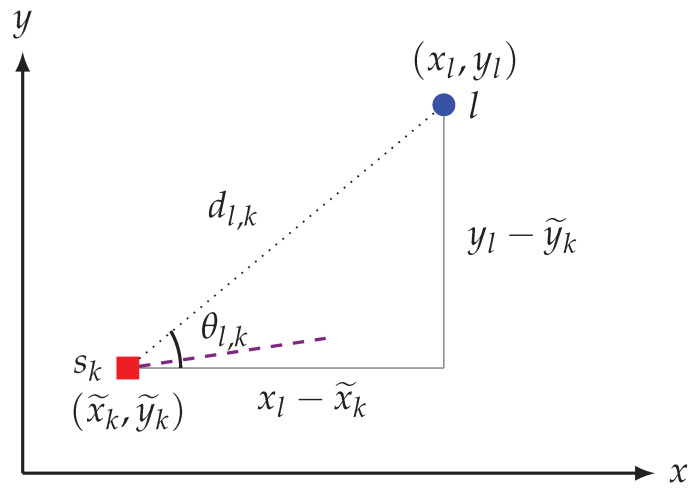
A simplified diagram illustrating the geometrical relation between the angle $\theta_{l,k}$, $s_{k}$, $d_{l,k}$ and the landmark position in the noiseless case. The dashed line in the plot indicates the robot heading direction, while the dotted line represents the sensor to landmark true distance.

**Figure 7 sensors-21-00155-f007:**
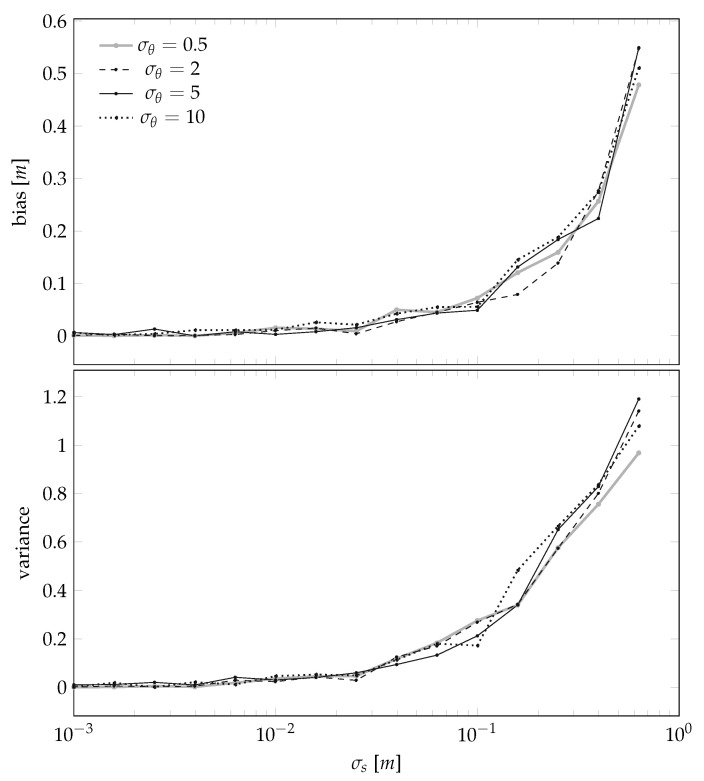
Bias and variance of the landmark position estimator of Section 3.1 as a function of the uncertainty in the sensor position evolution $\sigma_{s}$ for different sensor-to-landmark angle measurement standard deviations $\sigma_{\theta }$’s (in degrees) for the case L=1.

**Figure 8 sensors-21-00155-f008:**
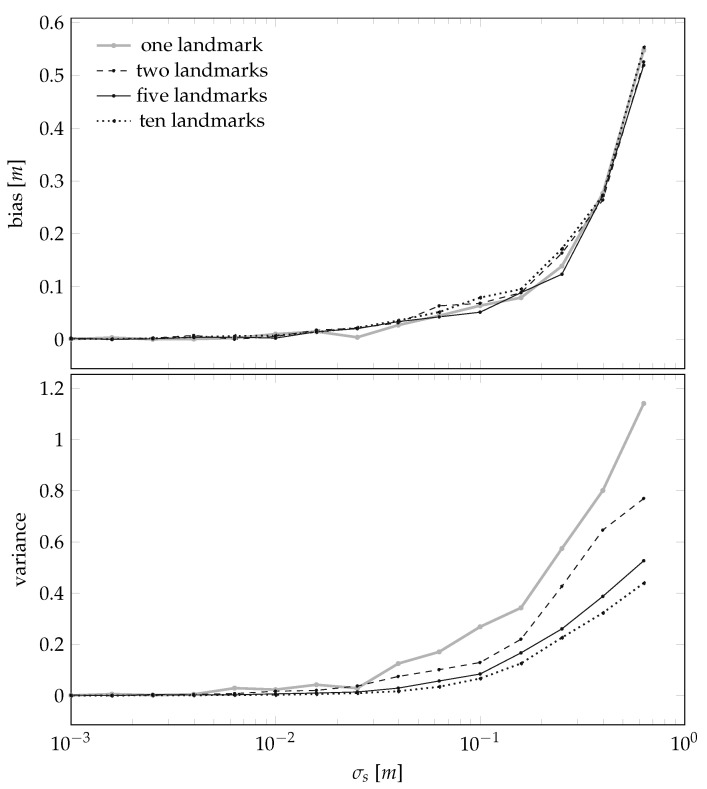
Bias and variance of the landmark position estimator of Section 3.1 as a function of the number of landmarks *L* for the case $\sigma_{\theta }=2$.

**Figure 9 sensors-21-00155-f009:**
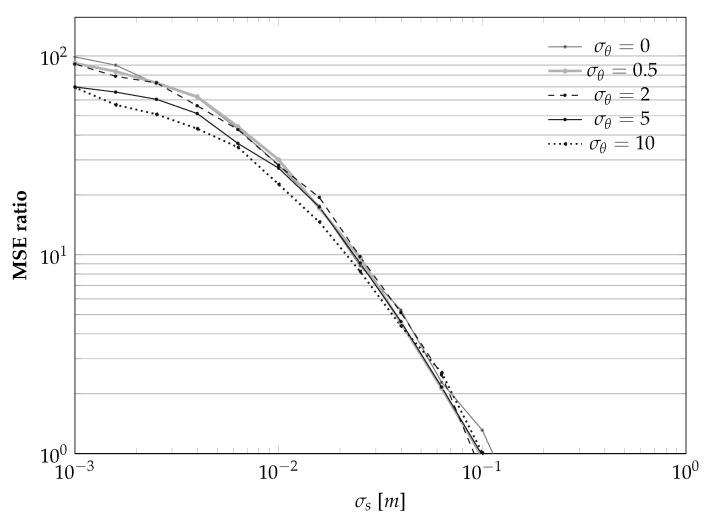
Dependency of the MSE ratio (Equation 31) on the standard deviation $\sigma_{\theta }$ of the sensor-to-landmark angle measurement error in (Equation 6), and on the standard deviation $\sigma_{s}$ of the sensor position evolution uncertainty in (Equation 8) for the case L=2.

**Figure 10 sensors-21-00155-f010:**
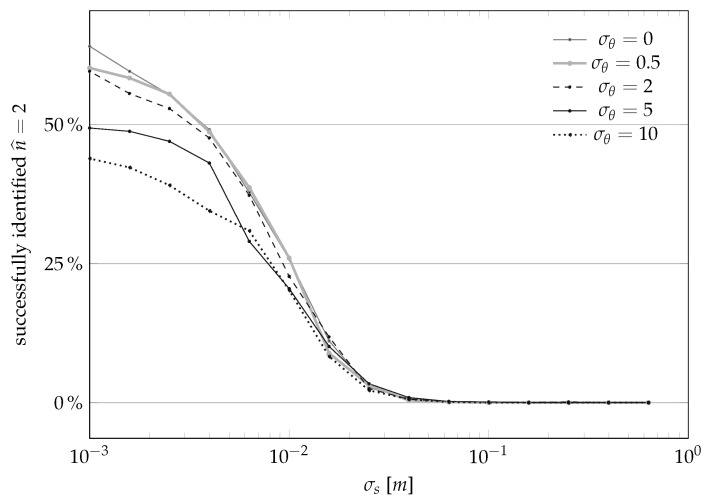
Summary of the dependency of the model order selection step on the standard deviations $\sigma_{\theta }$ and $\sigma_{s}$ for the case L=2. As the noises increase, the order selection process tends to select simpler models, as all the incorrectly classified model orders were of the kind $\hat{n}=1$.

**Figure 11 sensors-21-00155-f011:**
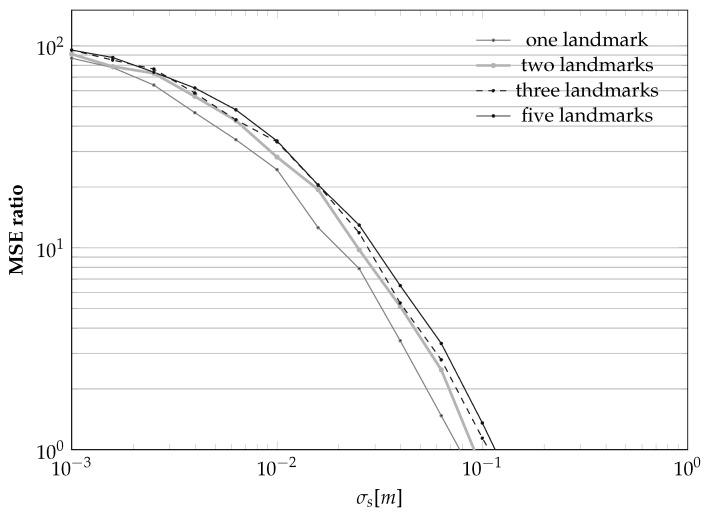
Dependency of the MSE ratio on the number of landmarks *L* as a function of the sensor position standard deviation $\sigma_{s}$ for the case $\sigma_{\theta }=2$. Decreasing $\sigma_{s}$ is, as expected, always beneficial, while increasing *L* is not so important.

**Figure 12 sensors-21-00155-f012:**
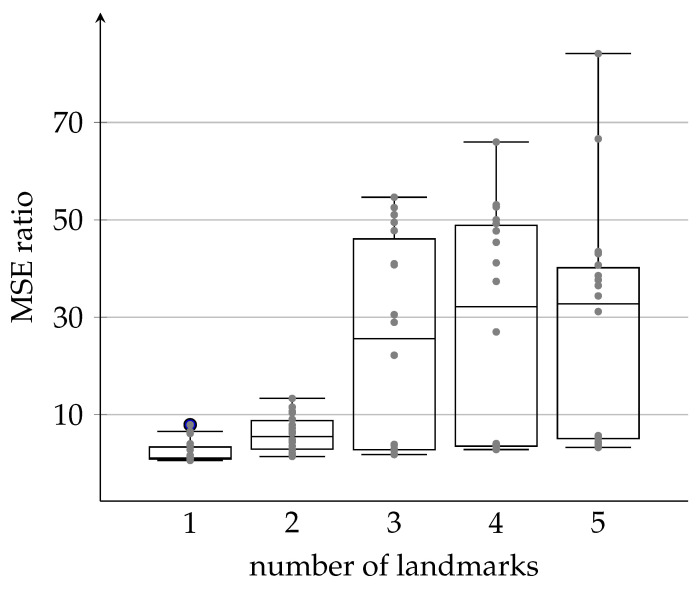
Statistics of the field tests for all possible combinations of datasets recorded in three different placements. Plots show clear increment of the MSE ratio with the increased number of the involved landmarks in the calibration process.

**Figure 13 sensors-21-00155-f013:**
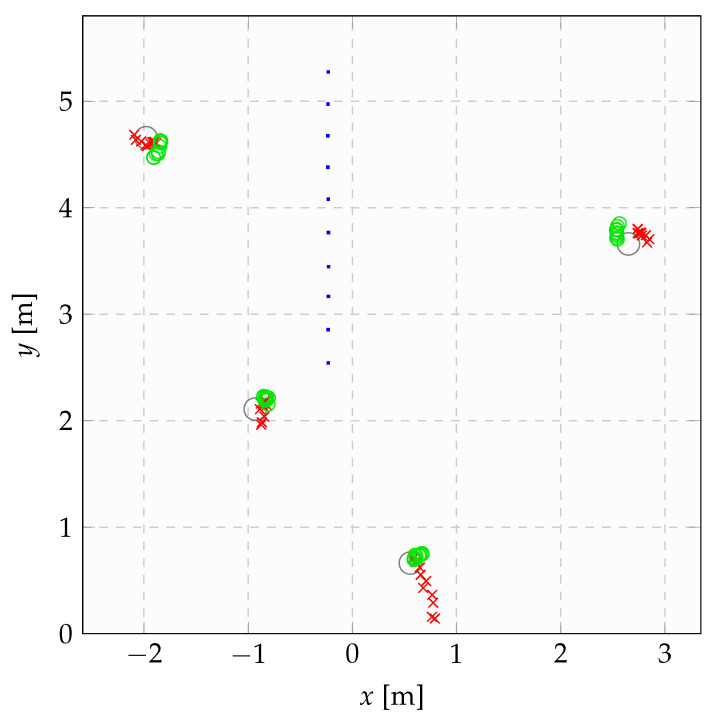
Example of the effects of the proposed calibration procedure on a field experiment. The actual sensor positions are plotted with a series of practically aligned blue dots, the true cylindrical landmarks in gray circles, the raw measurements taken by the sensor with red crosses, and the filtered distances, computed using the proposed strategy, in green circles. Ideally, the measurements should lie on the borders of the landmarks. It is immediately noticeable how the calibrated measurements lie much closer to such borders than the non-calibrated ones.

## Data Availability

The data presented in this study are available on request from the corresponding author. The data are not publicly available at the current time.

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
