# Peer review of "Calibrating Range Measurements of Lidars Using Fixed Landmarks in Unknown Positions"

_sensors, 2020, doi:10.3390/s21010155_

Round 1

Reviewer 1 Report

The work concerns the current topic of sensors dedicated to autonomous robots. The paper presents an algorithm that allows to improve the accuracy of indications of a "cheap" laser rangefinder. The introduction, problem analysis and research are carried out correctly. At work, I miss information about the lidar used, whether it was made by myself or bought, what parameters it had, etc., I suggest supplementing it.

 Summing up, I recommend this work for publication.

Reviewer 2 Report

The paper presents the work on the measurement on the positions of landmarks in unknown positions in an environment using a  sensor placed in ideally spaced positions. The method is based on triangulateralization: for each position of the sensor angles and distances are measured between the sensors and the landmarks and an algorithm calculates the estimated position. In the paper the theory is presented in details in a dedicated section. A section is dedicated to the numerical results and a field experiment validation.

It is not clear to me what happens when the sensors are not placed in ideally spaced positions, for example when the sensor follows a curved line or the steps are not equispaced due the rover mechanism. This contribution could not be considered as Gaussian dispersion with zero mean as the term in equation (8). At line 4 of pag 12 you write that the real sensor position is not available so you plug in the equation the expected sensor position.  The question is whether you could use an estimated position which different from the real one but also different from the expected equidistant position which would take into account a deviation from the equispaced hypothesis.

Moreover, since in the introduction you describe the measurement model having a bias and a dispersion, I propose to cite ISO/IEC GUIDE 98-3:2008, “Uncertainty of measurement — Part 3: Guide to the expression of uncertainty in measurement” called in the metrology community GUM. This would give the rigorous framework and reference to deal with uncertainty.

Figure 13 reports the results of the field experiment with 4 landmarks. it would be useful to present the picture of the setup, which is different from the one presented in Figure 4 with 5 landmarks.

Reviewer 3 Report

A method is proposed for calibration of 2D rangefinders based on stable landmarks. Its performance is elaborately analysed.

Sensor calibration has been very well studied. The use of a known environment is a very well-known way to calibrate sensors, though you seem to suggest this is an innovation in the presented work. E.g. in https://www.mdpi.com/2072-4292/11/8/905, section 4.1 and 4.2 a method is described to calibrate a laser rangefinder using walls in an indoor environment.

In your method you assume the locations of the sensor to be known. This takes away quite some flexibility in the recording of your range data. Why wouldn't you just take tape measurements between the landmarks and use trilateration to obtain their relative locations? You could then easily record measurements at large number of arbitrary sensor locations.

Your method assumes that the rotation of the scanner with respect to the adopted coordinate system is known. I doubt if that is a valid assumption as you don't know exactly how the lidar's internal coordinate system is aligned with the lidar box. This anyway needs to be better explained when setting up equation 9.

If I interpret equation 1 correctly, it reads "noisy range" = "bias" + "noise". I assume this should be "noisy range" = "true range" + "bias" + "noise". The same also holds for equation 3.

To properly calibrate the lidar, it would be advisable to have a large variety in the measured ranges. Is this sufficiently ensured in your protocol?

You first estimate the 2D positions of the landmarks and then calibrate the sensor. This seems inappropriate to me as the 2D landmark positions are then affected by the uncalibrated range measurements. Why don't you jointly estimate the landmark positions and range calibration parameters?

In figure 10 you want to show "the dependency of the model order selection step on the standard deviations". I don't see the results for different model orders in this figure.

You may assume that readers are familiar with least squares estimation.
